



# Near-real time detection of unexpected atmospheric events using Principal Component Analysis on the IASI radiances

Adrien Vu Van[1,2], Anne Boynard[1,2], Pascal Prunet[2], Dominique Jolivet[3], Olivier Lezeaux[2], Patrice Henry[4], Claude Camy-Peyret[5], Lieven Clarisse[6], Bruno Franco[6], Pierre Coheur[6] and Cathy Clerbaux[1,6]

[1]LATMOS/IPSL, Sorbonne Université, UVSQ, CNRS, Paris, 75005, France
[2]SPASCIA, Ramonville-Saint-Agne, 31520, France
[3]HYGEOS, Lille, 59000, France
[4]CNES (Centre National d'Etudes Spatiales), Toulouse, 31400, France
[5]IPSL, Institut Pierre-Simon Laplace, Paris, 75005, France
[6]Université libre de Bruxelles (ULB), Spectroscopy, Quantum Chemistry and Atmospheric Remote Sensing (SQUARES), Bruxelles, 1050, Belgium

*Correspondence to*: Anne Boynard (anne.boynard@latmos.ipsl.fr)

**Abstract.** The three IASI instruments on-board the Metop family of satellites have been sounding the atmospheric composition since 2006. More than 30 atmospheric gases can be measured from the IASI radiance spectra, allowing the improvement of weather forecasting, and the monitoring of atmospheric chemistry and climate variables.

The early detection of extreme events such as fires, pollution episodes, volcanic eruptions, or industrial releases is key to take safety measures to protect inhabitants and the environment in the impacted areas. With its near real time observations and good horizontal coverage, IASI can contribute to the series of monitoring systems for the systematic and continuous detection of exceptional atmospheric events, in order to support operational decisions.

In this paper, we describe a new approach for the near real time detection and characterization of unexpected events, which relies on the principal component analysis (PCA) of IASI radiance spectra. By analysing both the IASI raw and compressed spectra, we applied a PCA-granule based method on various past well documented extreme events such as volcanic eruptions, fires, anthropogenic pollutions and industrial accidents. We demonstrate that the method is well suited to detect spectral signatures for reactive and weak absorbing gases, even for sporadic events. Long-term records are also generated for fire and volcanic events, by analysing the available IASI/Metop-B data record.

## 1 Introduction

Atmospheric composition is changing fast locally, under natural and anthropogenic influences combined. Fire activity and local urban pollution are likely to increase in a warming climate (Hart, 2022). With their potential consequences on society and health, monitoring the events that impact atmospheric composition becomes more and more important.

Since the end of 2006 the IASI mission has been probing the troposphere from satellite to monitor the atmospheric composition globally, onboard of 3 successive Metop satellites (Clerbaux et al., 2009). Observation records and trends are available for several infrared absorbing species, such as methane ($CH_4$) (García et al., 2018), carbon monoxide (CO) (George et al., 2009), ammonia ($NH_3$) (Van Damme et al., 2021), ozone ($O_3$) (Dufour et al., 2018; Wespes et al., 2019) and dust (Capelle et al., 2014; Clarisse et al., 2019).  As the first goal of this mission is to feed meteorological



forecast using data assimilation, radiance Level 1C (L1C) data are received in near real time, around 2-3 hours after
the overpass of the satellite. This makes the detection of exceptional events possible, potentially right after they occur,
such as large biomass burning fires (Turquety et al., 2009; R'Honi et al., 2013), anthropogenic pollution episodes
(Boynard et al., 2014) or volcanic eruptions (Wright et al, 2022). With more than 1.2 million of radiance spectra per
instrument per day, the search for local extreme events in near real time is not straightforward. A limitation is also
associated with the lack of data when clouds are present in the field of view, as the retrieval algorithms fail to properly
derive atmospheric concentrations for trace gases. Cloudy data are hence filtered.
Soon after the launch of the first IASI instrument, it has been suggested to use the principal component analysis (PCA)
method to reduce data volumes by reconstructing the radiances using only the leading eigenvectors (Matricardi, 2010).
This compression not only allows to heavily decrease the data volume but also to ease the data dissemination. Now
available through the EUMETSAT (European organization for the exploitation of METeorological SATellites)
Advanced Retransmission Service (EARS-IASI), the PCA method allows meteorological centers to directly assimilate
the principal components (Collard et al., 2010; Matricardi et al., 2014; Guedj et al., 2015). It was also demonstrated
that using reconstructed IASI radiance results in a substantial reduction of random instrument noise for the analysis of
trace gases such as $NH_3$ or sulfur dioxide ($SO_2$) (Atkinson et al., 2010). However, it was decided to continue the
distribution of the entire radiance spectra (8461 spectral channels) as one of the concerns in the use of the PCA method,
for atmospheric chemistry studies, was the detection of spectral features associated with minor trace gases linked with
rare events in the reconstructed spectra. Examples are volcanic eruptions, which all differ in terms of gas and type of
ash emitted, and hence not enough representative cases were available in the training set. The same holds for biomass
burning fires releasing different amounts of specific species depending on the type of vegetation burned. With the
advent of the second and third IASI instrument together with the improvement of retrieval algorithms over time, a
number of short- and long-lived trace gases were identified in the IASI spectra above or downwind from strong
emission sources (Clarisse et al., 2011; De Longueville et al., 2021).
This paper describes the potential of the PCA applied on the IASI L1C (apodized radiance) data for the automatic,
near real time detection and characterization of exceptional events. The paper is organized as follows: Section 2
describes the IASI instrument and the dataset used in this study. Section 3 describes the PCA method. In Section 4, an
innovative approach based on the PCA method and IASI data granules is presented, which allows spectral
characterization of species in near real time. In Section 5, different case studies of exceptional past events are discussed,
such as volcanic, fire, and anthropogenic pollution episodes, along with industrial accidents, detected by IASI/Metop-
A and -B. Finally, conclusions are given in Section 6.
**2 The IASI radiance data**
IASI is a Fourier transform infrared spectrometer, which records the thermal infrared (TIR) radiation emitted by the
Earth and the atmosphere, between 645 cm$^{-1}$ and 2760 cm$^{-1}$, with 8461 channels sampled every 0.25 cm$^{-1}$ and a spectral
resolution of 0.5 cm$^{-1}$. An example of IASI spectrum along with the absorption band of several species is illustrated in
Fig. 1. Table A1 in Appendix A provides the selected spectral ranges defined for this work. Bands are identified with
HITRAN spectroscopic absorption parameters (Gordon et al., 2017, 2022) for infrared absorbing molecules. In this

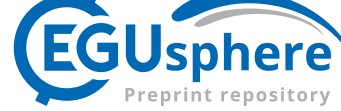

work IASI-A and IASI-B are used as a similar dataset. The IASI-A dataset is used for the study of events before the
launch of IASI-B and for creating the PCA training database (described here-after), and the IASI-B complete dataset
is used for data after 2013 to present. The two datasets have been shown to be highly consistent with no significant
drifts over time (García et al., 2016).
Each IASI instrument provides more of 1.2 million of spectra per day. IASI L1C data are disseminated by EUMETSAT
in 3-minute files (called "granule" hereafter) less than 3 hours after each overpass. Each granule contains 22 or 23
IASI scan lines with 120 pixels per line. With a wide swath width of ~2200 km, global observations are provided twice
a day, at 9:30 AM and 9:30 PM local time. IASI has an instantaneous field of view (FOV) at nadir with a spatial
resolution of 50 km x 50 km, composed of 2 x 2 circular pixels (IFOV), each corresponding to a 12 km diameter
footprint on the ground at nadir (Clerbaux et al. 2009).

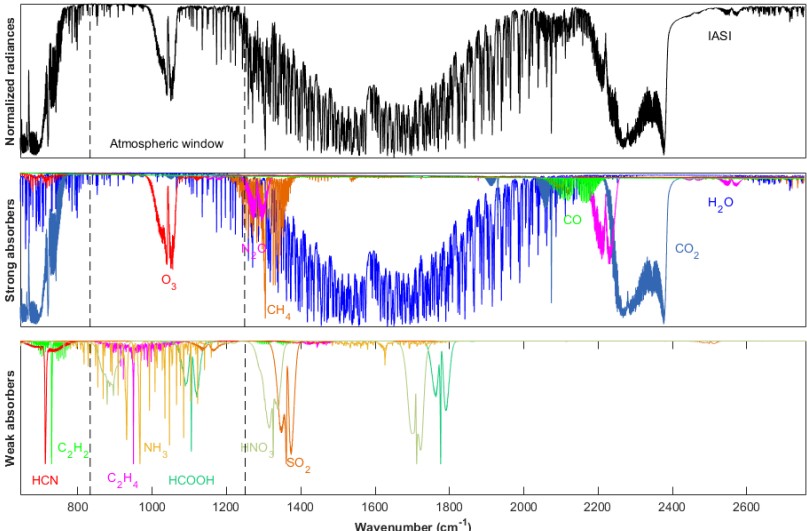


**Figure 1: Top panel: Example of IASI spectrum. Middle and bottom panels: radiative transfer simulations for the main and**
**weaker infrared absorbers, respectively.**
The atmospheric concentrations of some species are routinely retrieved from the spectral signatures (George et al.,
2009; Clarisse et al., 2011; Van Damme et al., 2013) and distributed through the AERIS database (iasi.aeris-data.fr).
Some exceptional events have been studied in details such as the 2010 Russian fires (R'honi et al., 2013), pollution in
the North China Plain (Boynard et al., 2014), and $SO_2$ anthropogenic pollution (Bauduin et al., 2014, 2016).
**3 The Principal Component Analysis Method**
**3.1 Basic concepts**
The PCA method for high spectral resolution sounders, such as IASI, is described in Atkinson et al. (2008). This
method is well suited to efficiently represent the limited amount of information contained in the 8641 IASI channels.



It relies on the use of a dataset of thousands of spectra representing the full range of atmospheric conditions from which
the principal components are calculated, the so-called "training database".
Let $\mathbf{Y}$ be a $m * n$ matrix representing a set of radiance spectra (where $m$ is the number of channels and $n$ is the number
of observations) and $\bar{y}$ its mean, the covariance matrix, $S_\epsilon$ ($m \times m$) of the noise-normalized spectra is given by:
$S_\epsilon = \frac{1}{n} \sum_{i=1}^{n} y_i N^{-1} y_i^T - \bar{y} N^{-1} \bar{y}^T$     (1)
where $N(m \times m)$ is the instrument noise covariance matrix. The PCA is based on the eigen decomposition of the
matrix $S_\epsilon$:
$S_\epsilon = E \lambda E^T$     (2)
where $\mathbf{E}$ is the matrix $m$ x $m$ of eigenvectors and $\lambda$ their associated eigenvalues.
The projection of a measured spectrum $y$ in the eigenspace $E$ is computed from:
$p = E^T N^{-1} (y - \bar{y})$     (3)
$p$ (dimension $m$) is the vector of the principal component scores, and is the representation of $y$ in the eigenspace.
Because most of the atmospheric signal is contained in a relatively small number of leading eigenvectors, with the
higher-rank eigenvectors containing mainly instrument noise, it is possible to discard the higher-rank eigenvectors,
and conservatively represent the spectrum in the eigenspace by a truncated vector of principal component scores, $p*$
of rank $m*$ ($m* < m$). $p*$ is thus a compressed representation of $y$. The reconstructed spectrum, $\tilde{y}$ (dimension $m$) is
given by:
$\tilde{y} = \bar{y} + N E^* p^*$     (4)
where $E^*$ is the matrix of the $m*$ first eigenvectors or principal components. We define the normalized residual vector
$r$ (dimension $m$) of the reconstruction by:
$r = N^{-1} (y - \tilde{y})$     (5)
By definition, if m* is taken equal to $m$, $\tilde{y} = y$ and the residual is the null vector. In nominal cases if the truncation
rank is carefully chosen, r essentially contains noise. Several techniques exist to estimate $m*$ the optimal number of
principal components in order to keep the essential part of the atmospheric signal and to remove the eigenvectors
containing mainly the measurement noise (e.g., Antonelli et al., (2004), Atkinson et al., (2010)).
**3.2 Construction of the training database**
For this study, around 120 000 IASI/Metop-A L1C spectra were selected during a full year (which was chosen as a
nominal year for avoiding excessive occurrence of extreme events such fires and volcanoes) on the global scale. The
database contains spectra associated with a good quality flag in order to only keep reliable data, acquired indifferently
during the day and the night, over land and sea, and regardless of the cloud cover. For each month of the year 2013



spectra were selected every five days (1, 6, 11, 16, 21 and 26 of each month). In order to not over-represent high
latitudes (due to the frequent overpasses over this area because of the polar orbiting), the following method was applied:
- between 90 and 75° only one spectrum is selected
- between 75 and 60°, two spectra are selected
- between 60 and 45°, three spectra are selected
- between 45 and 30°, four spectra are selected
- between 30 and 15°, five spectra are selected
- between 15 and 0°, six spectra are selected
Then a random selection was also applied to represent all the conditions of acquisition in terms of IASI scan lines and
IFOV.
**3.3 Number of eigenvectors, reconstruction score and "indicators"**
The eigen values quantify the explained variability of eigen vectors. These values are sorted in descending order. The
minimum number of eigenvectors needed to reproduce the signal in the raw radiances can be determined by analyzing
the magnitude of the eigenvalues. As shown in Fig. 2, around 20 principal components allow to depict most of the
atmospheric variability. However, as shown in Goldberg et al. (2003), keeping more than the first principal components
avoids a loss of information in the reconstructed spectra. In this study it is proposed to study the 3 IASI bands together
and choose the first 150 eigenvectors (over 8461) (Atkinson, 2010) to explain 99.99% of the atmospheric variability,
which allows to reduce the random noise without losing information in most of the reconstructed spectra.

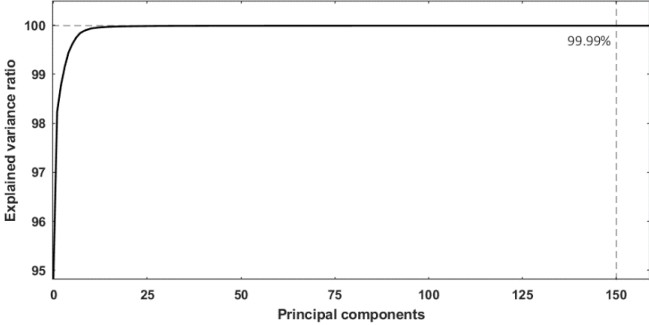


**Figure 2: Evolution of the explained variance ratio in function of the rank of principal components. 99.99% of the**
**atmospheric variability is depicted by the 150 first eigenvectors.**
**4 The IASI-PCA granule-extrema (GE) based method**
In order to use the maximum of information contained in the spectra for the different absorbing molecules, an original
method exploiting the full IASI spectral residual is developed. This approach, based on the IASI-PCA method, focuses





on intense spectral lines or narrow absorption bands of 0.5 cm$^{-1}$ – 2.0 cm$^{-1}$ width, depending on the molecular
absorption peaks (see Table A1 in Appendix A).

### 4.1 Granule maxima and minima

The near real time detection of exceptional events is performed on the IASI granule. The choice of applying the method
on the granule is convenient for the near real time aspect as it represents 3 minutes of IASI data which are received
every few 1-2 hours by the antenna. It allows not only a fast computing and statistical approach above a given
geographical region but also a preliminary approach allowing to characterize and localize outliers.
Each granule contains ~2700 radiance spectra, from which we compute the corresponding residuals. Indicators based
on the residuals statistics over the granule are calculated, such as minima, maxima, means, median and standard
deviations for each spectral channel. For a given granule, the largest positive and negative residual value for each
spectral channel is recorded in two arrays, called in the following "Granule Maxima" (GMA) and "Granule Minima"
(GMI) pseudo spectra. GMA and GMI pseudo spectra are associated with reconstruction errors of spectral emission
lines and absorption lines, respectively. The spectral vectors containing the mean of the spectral residuals (denoted $m$
in the following) and the standard deviation of the spectral residuals (denoted $\sigma$ in the following) over the granule are
also recorded.
**Figure 3 illustrates an example of GMA and GMI pseudo spectra for an intense fire event that occurred in Australia on 1**
**January 2020. The GMI pseudo spectrum (bottom panel) is characterized by detectable spectral features associated with a**
**poor reconstruction around 700, 950, 1100 or 2100 cm-1. Using spectroscopic database allows to associate some of these**
**strong peaks with contribution of different atmospheric components (see Section 5 for the identification of the molecules).**
**Similar spectral features can be seen in the GMA pseudo spectrum (top panel) albeit in emission and less intense.**

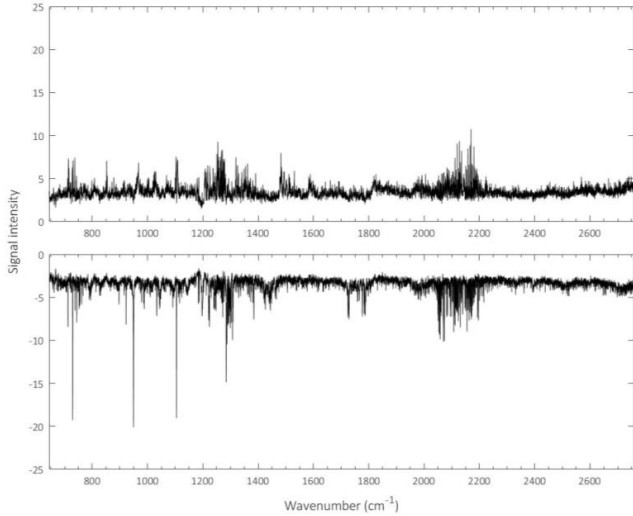


**Figure 3: Granule Maxima (GMA) (top) and Granule Minima (GMI) (bottom) pseudo spectra obtained from a granule of**
**IASI/Metop-B L1C data on 1 January 2020 over Australia.**



### 4.2 Detection thresholds

The identification of absorbing species is done using their spectroscopic properties. These parameters are provided from the HITRAN spectroscopic parameter database or in the literature if not available (see Table A1 in Appendix A). However, in order to apply the IASI-PCA-GE method for the systematic analysis of data of interest on the Metop-B time series, statistical thresholds are defined. Three different thresholds were defined, as described hereafter.

A dataset of 43 000 granules is created from random selection of granules taken on 1$^{st}$ day of each month between April 2013 and April 2021, containing both outlier and regular spectra during the Metop-B time series. Figure 4 shows the statistical distribution of the spectral extrema (in absolute values) calculated in the GMI and GMA pseudo spectra for 21 500 granules for both daytime and nighttime conditions. The blue box represents the 25$^{th}$ percentile, the median and the 75$^{th}$ percentile in the data. The upper and lower adjacent values are in black, and the red crosses respectively higher or lower than those values are considered as "outliers" in the dataset.

An absolute signal intensity for the 25$^{th}$ percentile is found around 5 (magenta dashed line) for all conditions. As a low filtering effect is expected from this threshold, in order to mitigate false negative detection even in case of low intensity events, the threshold $F_1 = 5$ is defined on the 25$^{th}$ percentile. Then, the extremum of each granule will be compared to $F_1$ to be selected or rejected by the algorithm.

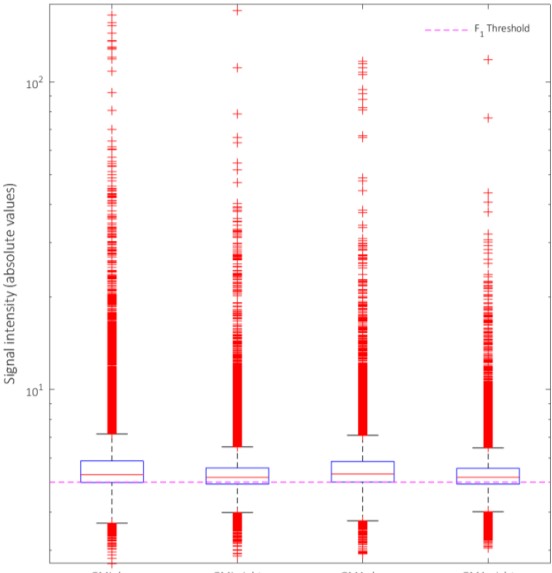

**Figure 4: Distribution of normalized GMI and GMA extrema in absolute values calculated from 21 500 granules both for day- and night-time conditions. The 25th percentile, the median and the 75th percentile are respectively represented at the bottom, middle and top of the blue box. The magenta dashed line also representing the 25th percentile of the signal intensity is around 5 for both conditions. The upper (top) and lower (bottom) adjacent values are represented in black.**



For a given granule selected using $F_1$, a second threshold $F_2$ is applied to select the channels associated with a poor
reconstruction in the GMI or GMA (depending on the atmospheric conditions). This threshold depends on the mean
and standard deviation spectral vectors, $m$ and $\sigma$ respectively, calculated over the granule. All IASI channels
characterized by a value (in absolute) larger than $F_2 = abs(m) + 1 * \sigma$ are selected.
Finally, a third threshold $F_3$ is empirically defined to reduce the noise without losing information. These thresholds are
defined based on the 21 500 GMI and 21 500 GMA pseudo spectra dataset previously introduced. The 99th percentile
value computed for each specific channel associated with a strong absorption of a molecule is selected as a $F_3$ threshold.
Table 1 provides the signal intensity thresholds for several species for daytime and nighttime conditions.
**Table 1: Signal intensity thresholds ($F_3$) for several species for day- and night-time conditions defined to reduce the noise**
**in GMA or GMI results.**

| Molecule | Channel (cm⁻¹) | GMI day | GMI night | GMA day | GMA night |
|----------|------------|---------|-----------|---------|-----------|
| HCN | 712.50 | -4.42 | -4.41 | 4.10 | 4.06 |
| C$_2$H$_2$ | 729.25 | -4.01 | -3.92 | 3.94 | 3.88 |
| C$_4$H$_4$O | 744 | -4.13 | -4.10 | 3.77 | 3.76 |
| HONO | 790 | -4.09 | -4.08 | 4.18 | 4.06 |
| NH$_3$ | 967 | -8.01 | -4.60 | 4.46 | 4.70 |
| C$_2$H$_4$ | 949 | -4.41 | -4.39 | 4.29 | 4.25 |
| CH$_3$OH | 1034 | -4.35 | -4.27 | 4.40 | 4.30 |
| HCOOH | 1105 | - 6.06 | -4.69 | 4.47 | 4.26 |
| HNO$_3$ | 1326 | -6.93 | -6.43 | 6.01 | 6.38 |
| SO$_2$ | 1345 | -7.52 | -4.92 | 4.38 | 4.46 |
| CO | 2111 | -6.89 | -4.72 | 4.58 | 4.28 |


**4.3 Towards a detection of extreme events in near real time**

Right after the reception of each IASI 3-minutes granule, the two GMA/GMI pseudo spectra are calculated as well as
other statistics of the residual over the granule. Then the three different thresholds defined in Section 4.2 are applied
to the GMA/GMI pseudo spectra in order to localize the pixels potentially associated with an event and the associated
channels. In case of anomalies (i.e., threshold overrun) in the GMA/GMI pseudo spectra, an alert is set-up along with
the targeted channels identified. the corresponding absorbing species are identified by associating the targeted channels
with their spectral range (see Appendix A), and the spatial distribution mapping of the detected pixels in the 3-minute
granule is produced. This allows to visualize and further study exceptional events. The IASI-PCA-GE method was
validated for past and documented events, four of which are described hereafter.  It is now running continuously,
delivering email alerts on a routine basis using the near real time IASI L1C radiance data. Most of these alerts are
associated with fires and volcanic eruptions.



**5 Case studies**
This section presents a demonstration of the IASI-PCA-GE method for several past extreme events. The method is
applied to IASI/Metop-A and the IASI/Metop-B L1C radiance data. Table 2 gives a brief description of the case studies
presented hereafter.
**Table 2:  Brief description of the four case studies analyzed in this section.**

| Type | Location | Date | AM/PM orbit | Instrument | Observed molecules |
|---|---|---|---|---|---|
| **Volcanic Eruption** | Ubinas/ Peru | 20/07/2019 | AM | IASI- B | $SO_2$, $HNO_3$ |
| **Fires** | Australia | 01/01/2020 | AM | IASI-B | HCN, $C_2H_2$, $C_2H_4$, HCOOH, CO, $NH_3$, $C_4H_4O$, $CH_3OH$ |
| **Anthropogenic pollution** | China | 13/01/2013 | PM | IASI-A | $NH_3$, $SO_2$, CO |
| **Industrial accident** | Iraq | 24/10/2016 | PM | IASI-B | $SO_2$, $HNO_3$ |


For each event, we identify the molecules in the outliers through analysis of the residual statistic, in order to identify
the spectroscopic feature characteristic of the studied gas, over a granule and applying the IASI-PCA-GE method. We
also provide distribution maps to illustrate the spatial distribution of the target event. When available, the maps are
compared to the existing retrieved IASI products ($NH_3$: Van Damme et al., 2021; $CH_3OH$ and HCOOH: Franco et al.,
2018; HCN: Rosanka et al., 2021; $SO_2$: Clarisse et al., 2012).
**5.1 Volcanic eruption events**
Volcanic eruptions have a major impact on atmospheric composition. $SO_2$, which has several strong absorption bands
in the TIR spectral range, is the most common molecule observed in the volcanic plume (Clarisse et al., 2012). Several
other species were previously observed by satellites in volcanic eruptions such as hydrochloric acid (HCl) (Clarisse et
al., 2020), hydrogen sulfide ($H_2S$) (Clarisse et al., 2011) and sulfuric acid ($H_2SO_4$) (Ackerman et al., 1994; Karagulian
et al., 2010), which can be injected in the stratosphere in case of high-altitude eruption (Rose et al., 2006; Millard et
al., 2006).
**5.1.1 The Ubinas (Peru) case study**
The IASI-PCA-GE method was applied to several volcanic eruptions. Here, we illustrate the findings for the eruption
in Ubinas, Peru on 20 July 2019 (Venzke et al., 2019). Instituto Geofísico del Perú (IGP) mentioned that seismic
activity suddenly increased during June 2019 and remained high during July 2019 with important ash emissions
causing the evacuation of the population in some areas affected by ashfall. Figure 5 illustrates the normalized GMI
pseudo spectrum obtained during this volcanic eruption corresponding to a granule taken in the area of the plume
during daytime.  A large difference between the reconstructed spectra and raw spectra is located in the $SO_2$ $v_3$ band
around 1371.5 and 1376 cm$^{-1}$ which is in agreement with results of Clarisse et al. (2008, 2012) showing the sensitivity



of the $v_3$ band. Indeed, the peak found at 1371.5 cm$^{-1}$ is associated with the presence of SO$_2$ plume in the upper
troposphere/lower stratosphere (~14 km, 150 hPa) between 0.5 DU and 200 DU (saturation) (Clarisse et al., 2011).
Such detection is expected in this case due to the high quantity of SO$_2$ emitted. It is worth noting that other peaks in
the GMI pseudo spectrum also show strong absorptions, which were associated with HNO$_3$. Even if this constituent
has previously been reported in volcanic plumes in some active degassing volcanoes (Mather et al., 2004), peaking in
the GMI at ~763 cm$^{-1}$, 879 cm$^{-1}$ and 897 cm$^{-1}$, and 1325 cm$^{-1}$ associated with $v_8$, $v_5$, $2v_9$, $v_3$ and $v_2$ nitric acid absorption
bands, respectively, it has never been observed by remote sensing before. As the analysis of the IASI HNO$_3$ L2 products
shows no HNO$_3$ enhancement, further investigations were performed to identify where the signature comes from.

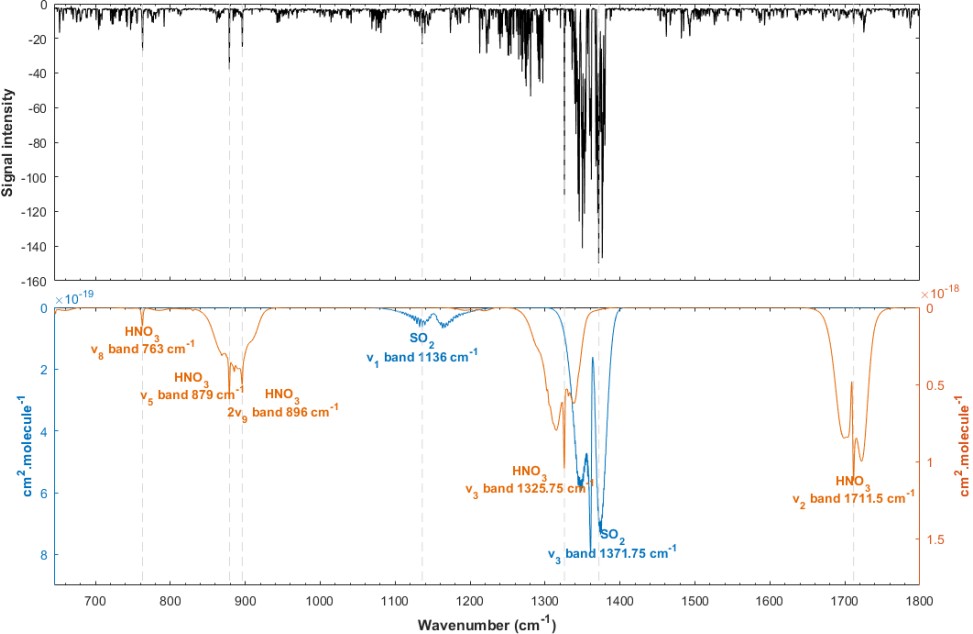


**Figure 5: Top: Example of GMI pseudo spectrum calculated from IASI/Metop-B L1C data during a volcanic eruption in**
**Ubinas, Peru on 20 July 2019 in the morning (AM orbit). Bottom: HITRAN spectroscopic parameters associated with the**
**absorption of HNO$_3$ and SO$_2$ are shown in blue and in orange, respectively.**
The HNO$_3$ detection by the IASI-PCA-GE method was further investigated by applying the whitening method
proposed by De Longueville et al. (2021). The use of a covariance matrix, calculated from a set of IASI spectra shows
similar results as those found with the IASI- IASI-PCA-GE method. However, using a covariance matrix excluding
the SO$_2$ absorption band, no HNO$_3$ spectral feature was found. This suggests that no acid nitric is present in the plume.
The features found in the HNO$_3$ absorption band by the IASI-PCA-GE method is likely related to SO$_2$ features given
that the SO$_2$ $v_3$ absorption band superimposes with the HNO$_3$ $v_3$ band.



Furthermore, other spectral signatures remain difficult to characterize in the $1200 - 1300$ cm$^{-1}$ spectral domain. This
spectral range corresponds to the absorption of different volcanic compounds such as ash, aerosols and other possible
volcanic molecules such as $H_2S$ or $H_2SO_4$ (Karagulian et al., 2010) but is also sensible to strong $H_2O$ absorptions.
After applying the thresholds filters defined in Section 4.2 to the GMI pseudo spectrum, the spatial distribution of the
pixels associated with outliers can be mapped. Figure 6 shows a plume of $SO_2$ (left) in Southeast America, with large
signal intensity values reaching around -150 in the center of the plume. The spatial distribution of the retrieved IASI
$SO_2$ L2 operational products (right) also shows the plume located in Southeast America and is in excellent agreement
with the $SO_2$ plume detected from the IASI-PCA-GE method.

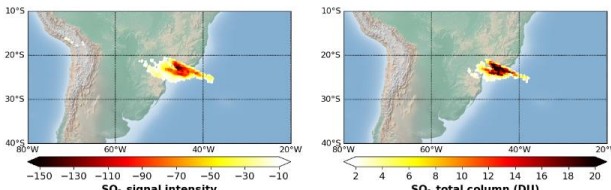


**Figure 6: Left: Spatial distribution of the residual values associated with SO₂ IASI-PCA-GE detections, using IASI/Metop-**
**B radiance data recorded on 20 July August 2019 in the morning (AM orbit). Right: SO₂ total column retrievals in Dobson**
**Units.**
**5.1.2 Volcanic eruption archive for IASI/Metop-B**
The time series of the $SO_2$ detections derived from the IASI-PCA-GE method is applied to IASI/Metop-B global
dataset over the 2013-2022 period. Figure 7 shows the comparison of the $SO_2$ IASI-PCA-GE signal intensity with the
$SO_2$ hyperspectral range indexes (HRI) product at 5 km (Bauduin et al., 2016). HRIs at 5 km are chosen because of a
good sensitivity around this altitude (Clarisse et al., 2014) compared to L2 $SO_2$ concentration data that are showing
concentration above 5 km (likely the high intensity volcanism). Only daily $SO_2$ extrema of both the IASI-PCA-GE
method and HRI product are compared. They are spatially co-located and associated with documented volcanic events
from the Global Volcanism Program, Smithsonian Institution (https://volcano.si.edu/). It is observed that both methods
are able to detect not only intense eruptions but also moderate or degassing volcanic events. The largest volcanic
eruptions detected during this period for both methods are Calbuco on 22 April 2015, Raikoke on 22 June 2019 and
Ubinas on 19 July 2019 (Sennert, 2015, 2019, 2019b). Furthermore, for all major events (corresponding to 2810 days
over 3373 days in total), an excellent correlation between HRI and IASI-PCA-GE signal intensity ($R^2 = 0.96$) is found
between the two datasets.



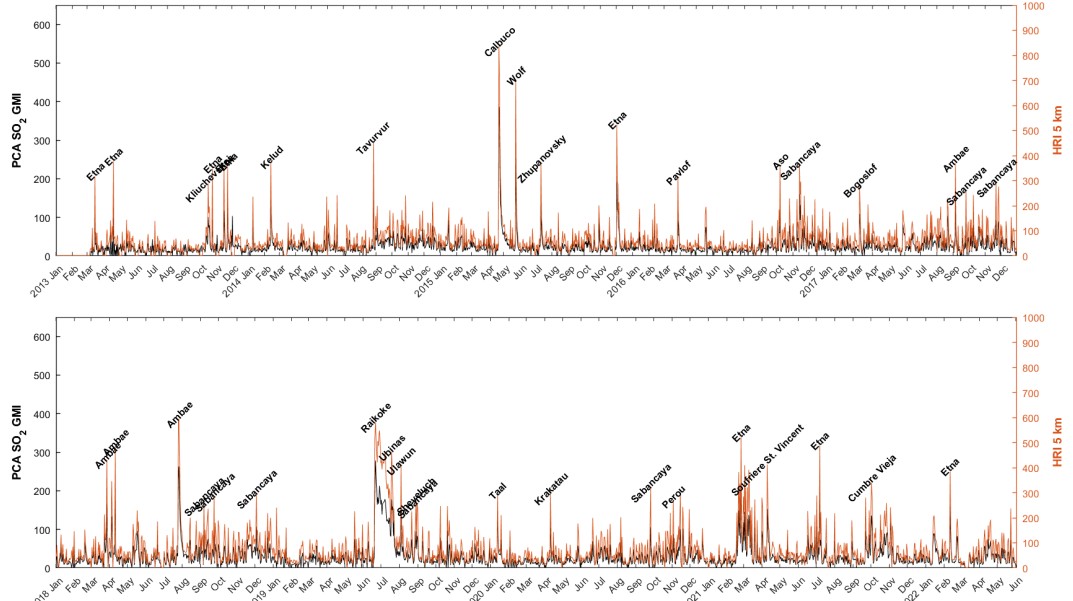

**Figure 7: Time series of SO₂ detections from IASI-PCA-GE method (grey) and the SO₂ HRI at 5 km (orange) based on the IASI/Metop-B L1C data for the 2013-2022 period. Only the daily extrema are shown in the time series.**

In order to analyse and understand the differences between the two records, the correlation between the latitudes of both datasets shown in Fig. 7 are plotted (see Figure 8). An excellent location correlation between both HRI and IASI-PCA-GE methods is observed for high intensity detections. However, some discrepancies are found in case of low intensity events, corresponding to commonly active degassing volcanoes.

Some specific latitudes associated with degassing volcanoes such as Sabancaya (Moussallam et al., 2017), the Vanuatu island arc with Ambae (Ani et al., 2012), Colima and Popocatepetl in Mexico (Varley et al., 2003) and the long eruptive Kilauea volcano (Garcia et al., 2021) respectively at 15.8° S, 15.4° S, 19.5° N, 19.0° N and 19.4° N are illustrated by the black horizontal and vertical dashed lines in Fig. 8. Furthermore, some daily maxima are located around 38° S, 37.5° N et 25.2° N and are respectively related to emissions from Copahue (Reath et al., 2019), Etna (Tamburello et al., 2013 ; Ganci et al., 2012) and several Chilean volcanoes.

The daily maxima located around 56° N are investigated and found to be associated with Kamchatka degassing volcanoes. Disperse latitudes of IASI-PCA-GE daily maxima are not consistent with the colocated HRI maxima. These differences between the IASI-PCA-GE and HRI methods can also be explained by the relation between plume altitude/temperature not represented in the principal components that will also affect the spectral reconstruction. As a result the location of daily maxima can be different in case of low intensity detections because of the PCA overestimation (or underestimation) of atmospheric anomalies. This also results on a non-linearity between retrieved concentrations and PCA intensities.



It is interesting to note that both IASI-PCA-GE and HRI detections observed at around 30° N and 65° N are associated
with anthropogenic emissions in the region of Sarcheshmeh Copper, one of largest industrial-mining complexes for
Copper that is emitting about 789.9 tons of $SO_2$ per day (Amirtaimoori et al., 2014) and over the Norilsk city, also well
know for its mining and smelting industries (Bauduin et al., 2016). That finding illustrates the capacity of both methods
to detect industrial emissions.

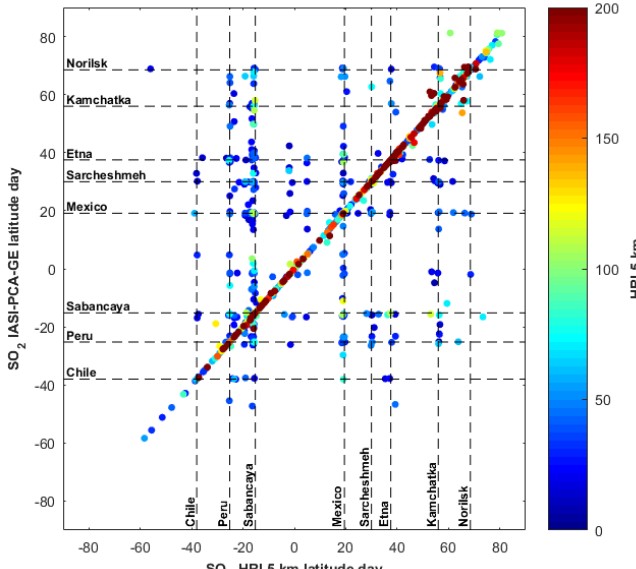


**Figure 8: Comparison of latitudes corresponding to the daily maxima detected for both IASI-PCA-GE $SO_2$ signal intensity**
**and HRI product between 2013 and 2022 with IASI-B L1C data during the day. The dashed lines show location**
**discrepancies.**
It is found that the relation between concentration and signal intensity is not linear and the PCA-based results cannot
be used for an accurate quantification of $SO_2$ concentrations. Indeed, IASI-PCA-GE signals will be dependent on the
molecule concentration but also on thermal contrast, and other surface parameters and atmospheric conditions. This
is why discrepancies are found at high latitudes between location of IASI-PCA-GE and HRI maxima, which are
associated with eruptions in the Kamchatka region.
**5.2 Fire events**
Fires can be a significant source of trace gases and aerosols in the atmosphere and several species were specifically
researched in fire events: CO, $NH_3$, formic acid (HCOOH), acetylene ($C_2H_2$), ethylene ($C_2H_4$), nitrous acid (HONO),
ethane ($C_2H_6$), acetonitrile ($CH_3CN$), methanol ($CH_3OH$), peroxyacetyl nitrate (CH3CO(OONO2)), hydrogen cyanide
(HCN), formaldehyde (HCHO), glyoxal (CHOCHO), and $CH_4$ (Li et al., 2000; Goode et al., 2000; Sharpe et al., 2004;
Coheur et al., 2009; Duflot et al., 2013; R'Honi et al., 2013; Zarzana et al., 2018, De longueville et al., 2021). The



IASI-PCA-GE method was applied to several case studies, but only one is presented here, selected during the fire
season occurring in Australia in 2019-2020.

### 5.2.1 The Australia case study

In Australia, fire events known as bushfires are occurring every year. Coupled with global warming and the lack of
rainfall in 2019-2020, the fires were particularly intense with burned areas covering more than 186 000 $km^2$. It was
shown that pyroconvection allowed the plume to reach the lower stratosphere around 15-16 km (Khaykin et al., 2020).
Many species were observed by ACE-FTS during that episode (e.g., Boone et al., 2020): CO, $C_2H_6$, $C_2H_2$, HCN,
HCOOH, $CH_3OH$, PAN, acetone ($CH_3COCH_3$) and $CH_3CN$.
The IASI-PCA-GE method was applied to the IASI/Metop-B L1C data on 1 January 2020. Figure 9 illustrates an
example of a normalized GMI pseudo spectrum obtained during the Australia fire event. As expected, peaks relative
to the CO absorption lines are found in the 2050-2200 $cm^{-1}$ spectral domain. Other peaks associated with the absorption
of molecules are also visible: HCN with a peak at 712.5 $cm^{-1}$, furan ($C_4H_4O$) at 744.5 $cm^{-1}$, $C_2H_2$ at 729.25 $cm^{-1}$, $C_2H_4$
at 949.5 $cm^{-1}$, HCOOH at 1105 $cm^{-1}$ and 1777 $cm^{-1}$, $CH_3OH$ at 1033.5 $cm^{-1}$, as well as peaks associated with $NH_3$ at
around 931 $cm^{-1}$ and 967 $cm^{-1}$.

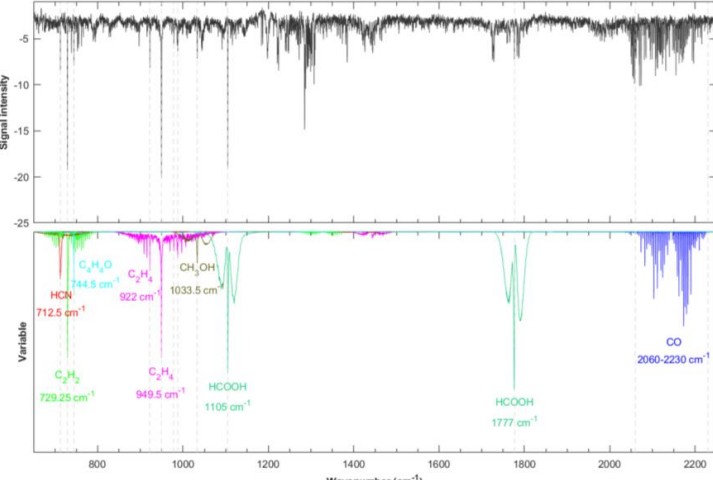


**Figure 9: Top: Example of GMI pseudo spectrum calculated from IASI/Metop-B L1C data during the intense fire event in**

**Australia on 1 January 2020 in the morning (AM orbit). Bottom: HITRAN spectroscopic parameter associated with the**

**absorption of different species are shown in colours.**

Figure 10 (left column) shows the spatial distribution of the residual values associated with the detected species in the
GMI pseudo spectrum. Despite their different lifetimes, the plumes for the different species are located in the same
region (around 180° E in the Pacific Ocean).
Carbon monoxide is retrieved in near real time (George et al., 2009) from IASI L1C and is used for monitoring fires
(Turquety et al., 2009). In Fig. 10, CO is both observed both with the IASI-PCA-GE and retrieval methods. However,



some discrepancies are found in terms of location and intensity. A few pixels are detected by the IASI-PCA-GE method
in the Southeast of Australia, which is in agreement with the CO operational L2 product. However, the retrieval method
is able to detect a larger plume over Australia compared to the IASI-PCA-GE method. Furthermore, a large plume is
also detected over the Pacific Ocean but is missed by the IASI-PCA-GE method. Note that, the high intensity CO
peaks are clearly detected in the residuals (c.f. Fig. 10). However, most of the missing pixels, in the PCA detection
results, are located above sea. That could be due to the combination between the database chosen in the PCA method
and the spectral domain high variability. Indeed, a higher thermal contrast variability is observed above land (Clerbaux
et al., 2009), but the database contains spectra representing the natural variability without differencing sea and land
pixels. As a result, the spectral reconstruction above sea with the PCA method will be less sensitive to spectral
variations, causing a reduced sensitivity above sea. Furthermore, the spectral region between 2050 and 2200 cm$^{-1}$ has
shown a large statistical distribution of extrema signals within the 21500 granules used for threshold calculation in
Section 4.2 allowing to set a restrictive threshold for the outlier detection for CO. That restrictive will also impact the
number of detected pixels. The sensitivity of PCA reconstruction outliers to strong CO concentrations in fires should
be more deeply investigated in further studies.
$NH_3$ is also retrieved in near real time (Van Damme et al., 2017) and observed in low concentration and occurrence
above Australia on the 1$^{st}$ January 2020 in the L2 retrievals and in low signal and occurrence in the IASI-PCA-GE
method. Some pixels are detected by the IASI-PCA-GE method but are not spatially correlated with the $NH_3$ total
column L2 data. A weak detection of $NH_3$ is expected since only low intensity peaks of $NH_3$ are found in the GMI
pseudo spectrum but two plumes are observed above both land and sea while L2 retrievals only show many isolated
pixels.
However, for other indicators the size of the plume differs: large plumes are found for $C_2H_2$, $C_2H_4$ and HCOOH while
smaller plumes are found for HCN, $C_4H_4O$ and $CH_3OH$. Those differences can be explained by the difference between
both methods. Indeed, the column maps includes the effects of radiative transfer (thermal contrast in particular), and
the presence of clouds can also induce differences between both products as the retrievals are highly sensitive to clouds.
For the IASI-PCA-GE method, the sensitivity for molecules detection highly depends on the selection of spectra to
construct the database and the thresholds chosen for the detection.

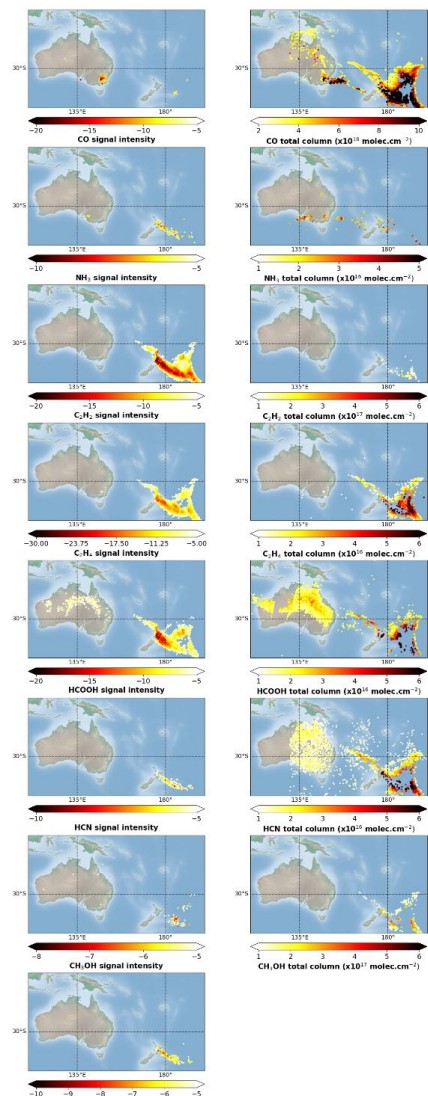

**Figure 10: Left: Spatial distribution of the residual values associated with CO, NH3, HCN, C₄H₂, C₂H₄, CH₃OH, HCOOH and C₄H₄O detections from IASI/Metop-B L1C data during the intense fire event in Australia on 1 January 2020 in the morning (AM orbit); right: same as left for the total column L2 data. There is no map of C₄H₄O total column L2 data because there is no retrieval available.**

**5.2.2 Fire archive for IASI/Metop-B**

Figure 11 illustrates the time series of the ethylene detections from IASI-PCA-GE method based on the IASI/Metop-B L1C data for the 2013-2022 period. $C_2H_4$ is a weak absorber often detected around 949 cm$^{-1}$ in case of high intensity fires and is able to show many high intensity peaks attributed to fire events. In the figure, the most intense fires are



characterized by their location (name indicated in black in Fig. 11). The presence of fires was validated by comparing
$C_2H_4$ detection to the IASI L2 CO that is shown to be a good fire tracker (Logan et al., 1981). The seasonality of fires
clearly appears during summer in the northern hemisphere mainly related to fires in Canada, Russia and Siberia and
during summer in the southern hemisphere with annual Australian and Indonesian fires. One of the largest detections
of the 2013-2022 period is associated with the 2019-2020 Australian bushfires discussed in section 5.2.1. Note that the
highest $C_2H_4$ intensity, observed on 29 July 2021 with a signal of 56, couldn't be associated with biomass burning as
no other indicators are present in the PCA-residuals. The source of this $C_2H_4$ enhancement is likely linked to
anthropogenic activities, as well as some other maxima, all located in Iran near the Irak border. This will be further
discussed in chapter 5.3.3.

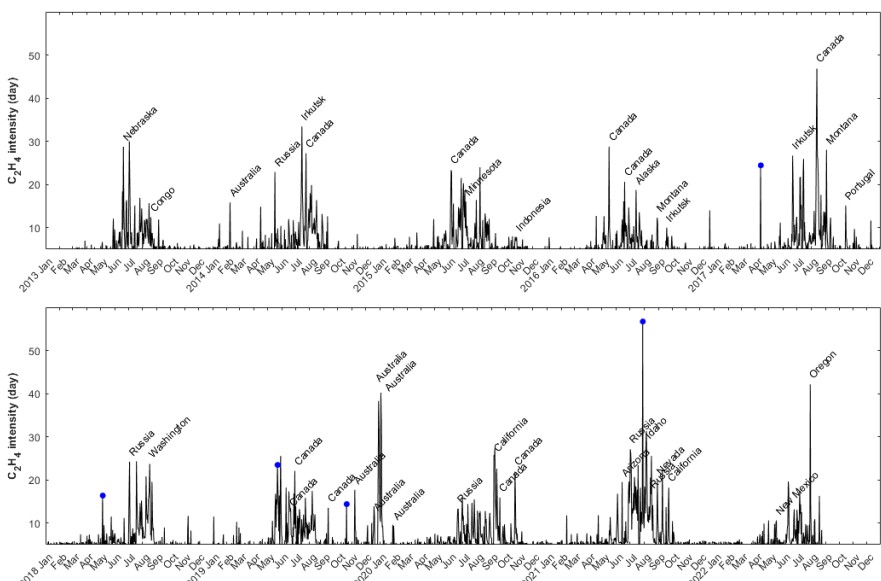


**Figure 11: Time series of $C_2H_4$ detections from IASI-PCA-GE method based on the IASI/Metop-B L1C data for the 2013-**
**2022 period. Only the daily extrema are shown in the time series. For clarity, the time series are separated into 2 periods:**
**2013-2017 (top panel) and 2018-2022 (bottom panel). Some events (blue dots) are associated with sporadic industrial**
**releases.**
**5.3 Anthropogenic pollution events**
**5.3.1 High pollution in China**
Boynard et al. (2014) investigated a severe pollution episode occurring in the North China Plain in January 2013. The
episode was caused by the presence of anthropogenic emissions combined with low wind speed and low altitude
boundary layer, which lead to weak mixing and dispersion of pollutants. The ability of IASI to detect high
concentrations of trace gases such as CO, $SO_2$, $NH_3$ as well as ammonium sulfate aerosol ($(NH_4)_2SO_4$) during nighttime
was demonstrated in case of large negative thermal contrast related to the winter season and the coal burning in China





for domestic heating. The IASI-PCA-GE method was applied on 13 January 2013 during nighttime. The normalized
GMA pseudo spectrum obtained during the China anthropogenic pollution is illustrated in Fig. 12. In order to optimize
the sensitivity of the method for a low intensity event, the $F_3$ thresholds were defined as $F_3 = 5$ for both day and
nighttime condition for the three species of interest (CO, $NH_3$ and $SO_2$). We clearly see a signal associated with CO,
$NH_3$, and $SO_2$ spectral emission, with the largest signal for $SO_2$ (value reaching ~18). The detection of $SO_2$ around
1345 cm$^{-1}$ is low compared to similar detection of $SO_2$ during volcanic eruptions. This result suggests that the $SO_2$
absorption bands around 1345 cm$^{-1}$ also allows the detection of $SO_2$ during anthropogenic pollution episodes, which
is in agreement with the finding of Bauduin et al. (2014, 2016). Finally, the spectral feature around 1180-1200 cm$^{-1}$
showing a low signal intensity is likely due to the IASI detector band 1 – band 2 inter-band domain that is well captured
in the IASI-PCA-GE method and should not be associated to an atmospheric absorption component.

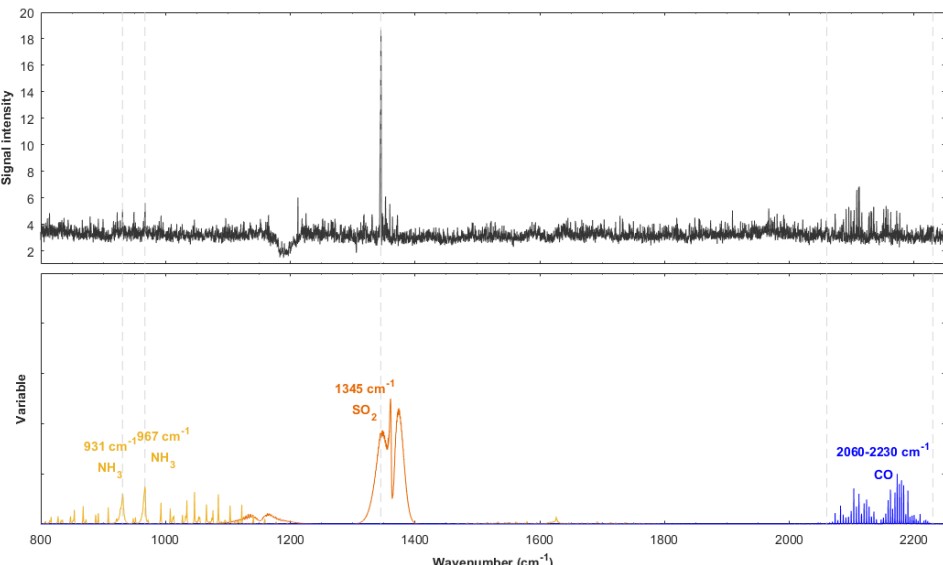


**Figure 12: Top: Example of GMA pseudo spectrum calculated from IASI/Metop-A L1C data during an anthropogenic**
**pollution event occurring in China on 13 January 2013 in the evening (PM orbit). Bottom: HITRAN spectroscopic**
**parameter associated with the absorption of different species are shown in colours.**
The spatial distribution of the residual values associated with the detected species in the GMA pseudo spectrum (see
Figure 12) is presented in Fig. 13 (left). The IASI-PCA-GE method allows the spectral detection of $NH_3$, $SO_2$, and CO.
However only a few pixels are detected for $NH_3$, which is due to the very low (<5) signal intensity found for that
species. We see the same behavior for CO. However, a clear $SO_2$ plume characterized by a signal reaching ~18 (at
1345 cm$^{-1}$ - see Fig. 12) is found by the IASI-PCA-GE method.
Figure 13 (right) illustrates the spatial distribution of $NH_3$ and CO total column and $SO_2$ plume altitude L2 data
retrieved from the IASI/Metop-A L1C data (Clarisse et al., 2012). The retrieval and IASI-PCA-GE methods shows





different patterns. We clearly see two plumes for SO$_2$ plume altitude and CO concentrations, but only few pixels of
detection are found for NH$_3$.

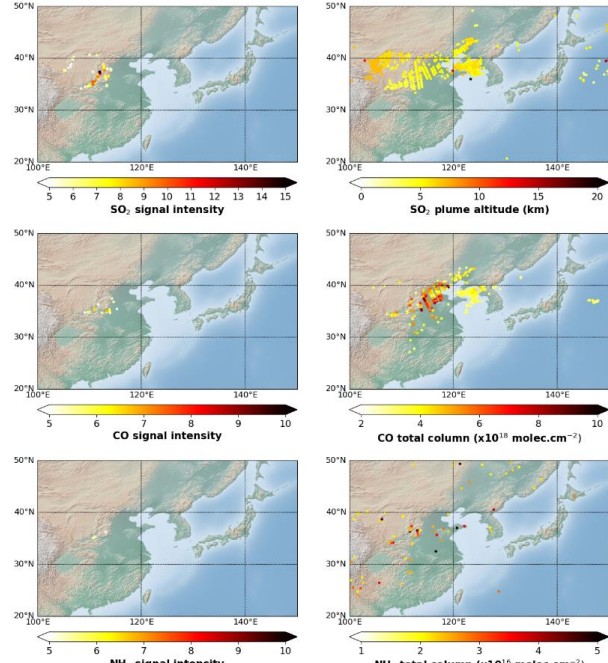

**Figure 13: Analysis of intense fire event in China on 13 January 2013 in the evening (PM orbit) based on IASI/Metop-A**
**L1C data. Left plots: spatial distribution of residual values associated with SO$_2$, CO and NH$_3$. Right plot: SO$_2$ plume altitude**
**retrievals (km), and CO and NH$_3$ total column retrievals (molec.cm$^{-2}$).**
**5.3.2 SO$_2$ released by a sulfur plant**
During the period extending from 20 October to 27 October 2016, a sulfur mine burnt in d'Al-Mishraq near Mosul,
Iraq. This fire on the sulfur plant, which was set by Islamic state, caused a large emission of SO$_2$ and other sulfured
species in the atmosphere, which was observed from several satellite instruments (Björnham et al., 2017). Similar plant
fires occurred in June 2003 during four weeks with approximately 600 kt of SO$_2$ emitted (Carn et al., 2004). This was
a major health hazard (Baird et al., 2012). Nearly thousand people were intoxicated due to toxic fire plumes, and two
Iraqis died.
Figure 14 illustrates the normalized GMI pseudo spectrum obtained during the Iraq industrial disaster on 24 October
2016 PM. The GMI pseudo spectrum is characterized by an absorption peak associated with HNO$_3$ artefact at ~1325.75
cm$^{-1}$ and two absorption peaks associated with SO$_2$ at 1345 cm$^{-1}$ and 1371 cm$^{-1}$. The signal intensity is about -14 for
SO$_2$ which suggests that the event is of a low-medium intensity. However, the SO$_2$ peaks found at 1371 cm$^{-1}$ and 1377
cm$^{-1}$ are mostly seen in case of intense volcanic eruptions, suggesting that the SO$_2$ concentrations are larger than



concentrations found above most of degassing volcanoes. This suggestion is well supported by Fig. 15 showing $SO_2$
total column up to 5 DU.
The presence of the artefact at 1325.75 $cm^{-1}$ can be explained by the reconstruction error in the 1300-1400 $cm^{-1}$ and
contribution of $SO_2$ and aerosols as observe in the case of Ubinas eruption (see section 5.1.1). As the event is less
intense than volcanic eruption, $HNO_3$ lower peaks around 750-950 $cm^{-1}$ are not observable.

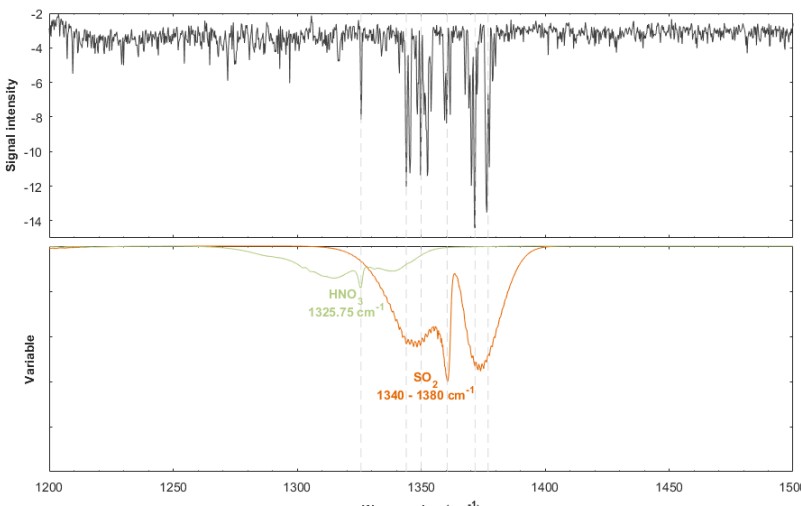


**Figure 14: Top: Example of GMI pseudo spectrum calculated from IASI/Metop-B L1C data during a sulphur plant fire**
**event occurring in Iraq on 24 October 2016 in the evening (PM orbit). Bottom: HITRAN spectroscopic parameter associated**
**with the absorption of different species are shown in colours.**
The spatial distribution of the residual values associated with $SO_2$ detections is illustrated in Fig. 15. The IASI-PCA-
GE method allows the spectral detection of this molecule in the region of interest four days after the fire started showing
the transport of the plume on the east part of the country. Less pixels are detected by the IASI-PCA-GE method than
by the L2 retrieval method. This can be explained by the fact that $SO_2$ thresholds associated with the IASI-PCA-GE
method were empirically chosen to minimize false positive detections, and thus the detections of low intensity residuals
can be missed.



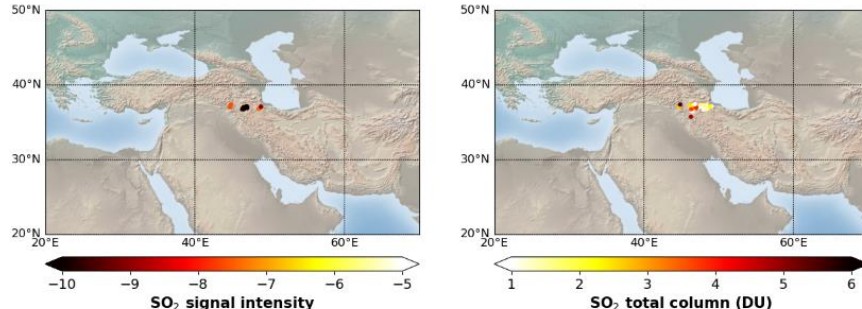


**Figure 15: Analysis of sulphur plant fire event in Iraq on 24 October 2016 in the evening (PM orbit) based on IASI/Metop-A L1C data. Left plot: spatial distribution of residual values associated with SO₂. Right plot: SO₂ total column in Dobson Unit.**

### 5.3.3 $C_2H_4$ sporadic emission at the border of Iran/Irak

In Section 5.2.2 we reported that the IASI-PCA-GE method is well suited to detect biomass burning by using the $C_2H_4$ indicator, found in conjunction with other signatures of molecules usually associated with fire activity. Among the events that we inventoried, on a few occasions, we found intense signatures in the Iran/Iraq region with no other absorption than $C_2H_4$, which suggests that sources other than biomass burning – likely due to anthropogenic activities – are at play. The main event occurred in July 2021 and some other weaker ones are also identified in Fig. 11. By averaging IASI data over time and using a supersampling technique, Franco et al. (2022) exposed and identified over 300 worldwide emitters of $C_2H_4$, emanating from petrochemical clusters, steel plants, coal-related industries, and megacities. However, no $C_2H_4$ point source was formally identified in this Iran/Iraq region. But the method described in this paper is well suited to also detect sporadic events , which contrasts with the continuous emissions exposed by Franco et al. (2022). Indeed, oversampling methods are well suited for the detection of regular, even weak, anthropogenic sources, but typically miss transient sources lasting for less than 24 hours. A new analysis was therefore performed on the events spotted by the IASI-PCA-GE method, which led to the identification of plumes lasting for only a few hours (see Figure 16), for specific days as identified on Fig. 11. Although visible satellite imagery and independent online information indicate the presence of oil and gas activities in that area, no firm identification was possible, and further investigation is needed to identify the potential sources of these sporadic emissions.



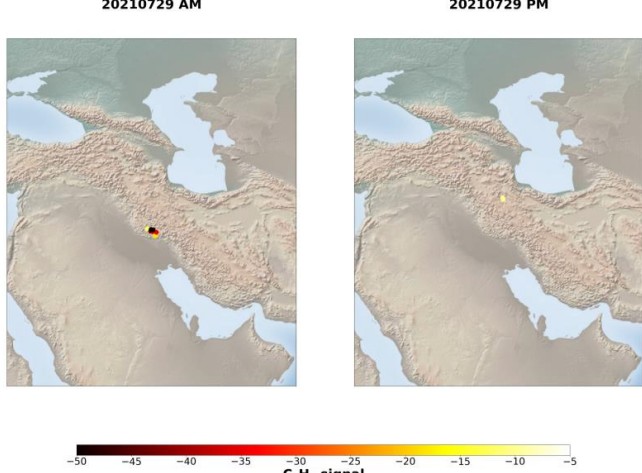

**Figure 16: Analysis of acetylene sporadic emission event in Iraq on 29 July 2021 based on IASI/Metop-A L1C data. Left plot: spatial distribution of residual values associated with $C_2H_4$ during the morning orbit. Right plot: spatial distribution of residual values associated with $C_2H_4$ during the evening orbit.**

**6 Conclusions and perspectives**

This paper presents an innovative approach, based on a PCA method applied on the IASI radiance, allowing the detection and characterization of exceptional events in near real time. This new method, the IASI-PCA granule extrema (GE) method, consists in focusing on extrema calculated within a geographical region. A statistical selection is made for focusing on abnormal variability in IASI channels (detection of outliers) in order to detect the contribution of specific molecules from different types of events. It is applied to the three-minute granule of IASI observations allowing the near real time detection of a series of short-lived trace gases.

Using a dataset representing the full range of atmospheric conditions, we show that the PCA method is well suited to efficiently detect outliers. The analysis of the outliers allows the identification of spectral features exceeding the natural variability of several absorbing species especially for weak absorbers, emitted during fires, volcanic, anthropogenic pollution, or industrial disaster. The method is more robust than previous retrieval methods when the spectra are cloud-contaminated.

The analysis of several case studies shows a good sensitivity of the IASI-PCA-GE method, which is able to detect weak absorbers such as $SO_2$, HCN, $C_2H_2$, $C_2H_4$, $CH_3OH$, $C_4H_4O$ and $NH_3$. We also showed that the method is well suited to detect transient events that last only a few hours/days.

Our work shows that within a granule the negative part of residuals (GMI) contains more information than the positive represented by the GMA. However, the latter contains relevant information in case of negative thermal contrasts, allowing the detection of specific events such as the recurrent anthropogenic pollution events occurring in China in winter.



The IASI-PCA-GE method is better suited to detect short-lived species and only species characterized by spectral
absorption lines are defined in the method. Species such as PAN, $CH_3COOH$ and $CH_3COCH_3$ characterized by a
broadband absorption are more difficult to detect with the IASI-PCA-GE method. Less conclusive results were
obtained for CO which has a lower atmospheric concentration variability because of its longer lifetime. This could also
be related to the training database used in this study, which still contains outliers impacting the sensitivity. In a next
version, the method will be optimized by removing the outliers still present in the database. Furthermore, as the PCA
detection depends on different geophysical parameters such as the temperature, thermal contrast or the construction of
the principal component basis, the detection of most extreme cases of concentration anomalies is possible but the
sensitivity of the method will depend on the latitude of observations. Added to that, in case of intense volcanic eruption,
some signal artefacts can appear due to the high reconstruction error observed in the 1300-1400 $cm^{-1}$.
Finally, this paper shows the capacity of PCA detection to detect different species from an event to another, especially
in case of fire events, which suggest the possibility to categorize fire events based on judicious combinations of species.
The method also proves useful to derive consistent records for fire and volcanic events, and data will continue to
accumulate over time as the method is now routinely implemented. Further work is still needed to avoid false
detections, such as those associated with $HNO_3$ which are due to the correlation between different absorption bands
for the same molecule, likely interfering with $SO_2$ present in the plumes.
**Data availability statement**
The IASI L2 $SO_2$ ($SO_2$, $NH_3$, CO, HCOOH data can be downloaded from the Aeris portal https://iasi.aeris-data.fr/SO2/
(https://doi.org/10.25326/42); https://iasi.aeris-data.fr/nh3/ (https://doi.org/10.25326/10); https://iasi.aeris-data.fr/CO/
(https://doi.org/10.25326/64); https://iasi.aeris-data.fr/hcooh/ (https://doi.org/10.25326/15)). The VOC retrievals
($CH_3OH$ and HCOOH: https://doi.org/10.1029/2018JD029633; HCN: https://doi.org/10.5194/acp-21-11257-2021) are
processed by Franco Bruno at ULB (bruno.franco@ulb.be).
**Acknowledgments**
A. Vu Van acknowledges funding from SPACIA SA through an ANRT CIFRE PhD grant. IASI is a joint mission of
EUMETSAT and the Centre National d'Etudes Spatiales (CNES, France). The IASI Level 1C data are distributed in
near real time by EUMETSAT through the EUMETCast system distribution. The authors acknowledge the AERIS
data infrastructure (https://www.aeris-data.fr) for providing access to the IASI Level 1 radiance and Level 2
concentration data used in this study, and CNES for financial support.







**Appendix**
**Table A1: Infrared absorption spectral ranges used for the detection of molecular absorptions.**

| Computed spectral range ($cm^{-1}$) | Width ($cm^{-1}$) | Molecule | Infrared absorption peaks | References |
|---|---|---|---|---|
| 711.50 - 713.50 | 2.00 | HCN | 712.5 $cm^{-1}$ (Q-branch) | (Gordon et al., 2017, 2022) |
| 729.25 - 730.00 | 0.75 | $C_2H_2$ | 729.25 $cm^{-1}$ (Q-branch $v_5$ band) | (Gordon et al., 2017, 2022) |
| 744.25 – 744.75 | 0.50 | $C_4H_4O$ | 744.5 $cm^{-1}$ ($v_{20}$ band) | (Gordon et al., 2017, 2022) |
| 763.00 - 763.75 | 0.75 | $HNO_3$ | 763 $cm^{-1}$ (Q-branch $v_8$ band) | (Gordon et al., 2017, 2022) |
| 790.25 – 790.75 | 0.50 | HONO | 790 cm-1 (Q-branch trans-$v_4$ band) | (Barney et al., 2000) |
| 853.50 - 854.25 | 0.75 | $NH_3$ | 854 $cm^{-1}$ (Q-branch) | (Gordon et al., 2017, 2022) |
| 867.75 - 868.75 | 1.00 | $NH_3$ | 868 $cm^{-1}$ (Q-branch) | (Gordon et al., 2017, 2022) |
| 878.50 - 880.00 | 1.50 | $HNO_3$ | 879 $cm^{-1}$ (Q-branch $v_5$ band) | (Gordon et al., 2017, 2022) |
| 887.25 - 888.25 | 1.00 | $NH_3$ | 888 $cm^{-1}$ (Q-branch) | (Gordon et al., 2017, 2022) |
| 891.75 - 892.25 | 0.50 | $NH_3$ | 892 $cm^{-1}$ (Q-branch) | (Gordon et al., 2017, 2022) |
| 895.50 - 896.75 | 1.25 | $HNO_3$ | 896 $cm^{-1}$ (Q-branch $2v_9$ band) | (Gordon et al., 2017, 2022) |
| 908.00 - 909.00 | 1.00 | $NH_3$ | 908.25 $cm^{-1}$ (Q-branch) | (Gordon et al., 2017, 2022) |
| 931.75 - 933.75 | 2.00 | $NH_3$ | 930 $cm^{-1}$ (Q-branch) | (Gordon et al., 2017, 2022) |
| 949.00 - 950.50 | 1.50 | $C_2H_4$ | 949 $cm^{-1}$ (Q-branch $v_7$ band) | (Gordon et al., 2017, 2022) |
| 966.00 - 968.00 | 2.00 | $NH_3$ | 967 $cm^{-1}$ (Q-branch) | (Gordon et al., 2017, 2022) |
| 991.75 - 993.50 | 1.75 | $NH_3$ | 992.75 $cm^{-1}$ (Q-branch) | (Gordon et al., 2017, 2022) |
| 1007.75 - 1008.25 | 0.50 | $NH_3$ | 1008 $cm^{-1}$ (Q-branch) | (Gordon et al., 2017, 2022) |
| 1033.00 - 1033.75 | 0.75 | $CH_3OH$ | 1033.5 $cm^{-1}$ (Q-branch) | (Razavi et al., 2011) |
| 1034.00 - 1034.25 | 0.25 | $NH_3$ | 1034 $cm^{-1}$ (Q-branch) | (Gordon et al., 2017, 2022) |
| 1046.25 - 1047.25 | 1.00 | $NH_3$ | 1047 $cm^{-1}$ (Q-branch) | (Gordon et al., 2017, 2022) |
| 1065.75 - 1066.25 | 0.50 | $NH_3$ | 1066 $cm^{-1}$ (Q-branch) | (Gordon et al., 2017, 2022) |
| 1075.75 - 1076.25 | 0.50 | $NH_3$ | 1076 $cm^{-1}$ (Q-branch) | (Gordon et al., 2017, 2022) |
| 1084.50 - 1085.75 | 1.25 | $NH_3$ | 1085 $cm^{-1}$ (Q-branch) | (Gordon et al., 2017, 2022) |
| 1103.00 - 1104.25 | 1.25 | $NH_3$ | 1104 $cm^{-1}$ (Q-branch) | (Gordon et al., 2017, 2022) |
| 1104.50 - 1105.75 | 1.25 | HCOOH | 1105 cm-1 (Q-branch $v_6$ band) | (Gordon et al., 2017, 2022) |
| 1121.50 - 1122.75 | 1.25 | $NH_3$ | 1122 $cm^{-1}$ (Q-branch) | (Gordon et al., 2017, 2022) |
| 1325.75 - 1326.25 | 0.50 | $HNO_3$ | 1326 $cm^{-1}$ ($v_3$ band) | (Gordon et al., 2017, 2022) |
| 1344.50 - 1346.50 | 1.00 | $SO_2$ | 1345 $cm^{-1}$ ($v_3$ band) | (Gordon et al., 2017, 2022) |
| 1370.50 - 1372.00 | 1.50 | $SO_2$ | 1371 $cm^{-1}$ ($v_3$ band) | (Gordon et al., 2017, 2022) |
| 1375.75 - 1377.00 | 1.25 | $SO_2$ | 1376 $cm^{-1}$ ($v_3$ band) | (Gordon et al., 2017, 2022) |
| 1710.75 - 1711.50 | 0.75 | $HNO_3$ | 1711 $cm^{-1}$ (Q-branch) | (Gordon et al., 2017, 2022) |
| 1776.75 - 1777.25 | 0.50 | HCOOH | 1777 cm-1 ($v_3$ band) | (Gordon et al., 2017, 2022) |
| 2111.00 – 2112.25 | 1.25 | CO | 2111.50 $cm^{-1}$ (P-branch) | (Hadji-Lazaro, 1999) |
| 2123.00 – 2124.25 | 1.25 | CO | 2123.75 $cm^{-1}$ (P-branch) | (Hadji-Lazaro 1999) |
| 2130.00 – 2132.25 | 2.25 | CO | 2131.75 $cm^{-1}$ (P-branch) | (Hadji-Lazaro, 1999) |
| 2157.75 – 2158.75 | 1.00 | CO | 2158.00 $cm^{-1}$ (R-branch) | (Hadji-Lazaro, 1999) |
| 2164.75 – 2166.00 | 1.25 | CO | 2165.75 $cm^{-1}$ (R-branch) | (Hadji-Lazaro, 1999) |




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
