# Peer review of "Near-real time detection of unexpected atmospheric events using"

_EGUsphere, 2022_

## Author Comment (AC1)

**Response to Referee #2**

**The authors would like to thank the referee for her/his review. Below are our responses to the comments brought up by the referee. Referee's comments and our replies are marked in blue and in black, respectively. In italic are the changes made in the manuscript.**

General comments

The paper describes a method for detection of unusual IASI spectra based on the noise normalised (PC) reconstruction residual. Each channel of the noise normalised reconstruction residual is a linear function of the radiances. 11 channels corresponding to absorption lines of selected gases are chosen for the detection (in the appendix a larger set of channels (ranges) is presented but the relation and role of each of these channel sets is not clear). While, by construction, the method is well suited to identify unusual spectra, the allocation to specific molecules can cause false detections as illustrated in the paper with the (false) HNO3 detection in a volcanic plume.

The set of 11 spectral intervals corresponding to absorption lines of selected gases were defined to identify the potential presence of molecules involved in an exceptional event. However in this work, only one spectral channel per molecule associated with the strongest absorption peak was used for the detection. For clarity concern, we added in Table 1 the spectral ranges and the infrared absorption peaks associated with the 11 species and we removed Table A1.

Specific comments

Section 3.1 needs some corrections. N is defined as the instrument noise covariance matrix, which is consistent with its use in equation 1, but not in the following equations, where it should be the matrix square root of the instrument noise. Line 102: this is not a projection (look up the definition). Line 107: "conservatively"?? Line 115: "optimal number" optimal in what sense? Line 97: is it really necessary to give a formula for the covariance matrix, especially since this formula is not the best way to actually compute it.

Section 3.1 has been modified in order to correct and improve the description of the methodology. Section 3.1 of the revised manuscript is as follows:

*3.1 Basic concepts*

*The PCA method for high spectral resolution sounders, such as IASI, is described in Atkinson et al. (2008). This method is well suited to efficiently represent the amount of information contained in the 8641 IASI channels. It relies on the use of a dataset of thousands of spectra representing the full range of atmospheric conditions from which the principal components are calculated, the so-called "training database".*

*One considers an ensemble Y of n IASI radiance spectra $\mathbf{y}$ of dimension m (where m is the number of channels and n is the number of observations). Let denote $\mathbf{N}^{-1}\overline{\mathbf{y}}$ the mean and $\mathbf{S}_\epsilon$ (m × m) the covariance of the normalized ensemble of spectra $\mathbf{N}^{-1}Y$. $\mathbf{N}$ is the noise normalisation matrix and is defined as the square root of $\mathbf{S}_y$ (m × m) the instrument noise covariance matrix associated to the IASI spectra.*

*The PCA is based on the eigen decomposition of the matrix $\mathbf{S}_\epsilon$ :*

$$\mathbf{S}_\epsilon = \mathbf{E}\,\Lambda\,\mathbf{E}^T \qquad (1)$$

*where $\mathbf{E}$ is the matrix m x m of eigenvectors and $\Lambda$ the diagonal matrix of their associated eigenvalues. The representation of a measured spectrum $\mathbf{y}$ in the eigenspace $\mathbf{E}$ is obtained by:*

$$\mathbf{p} = \mathbf{E}^T \mathbf{N}^{-1}(\mathbf{y} - \overline{\mathbf{y}}) \qquad (2)$$

*$\mathbf{p}$ (dimension m) is the vector of the principal component scores.*

*The analysis consists in representing the multidimensional IASI spectra in a lower dimensional space, which accounts for most of the variance seen in the data. This space is spanned by a truncated set of the eigenvectors of the data covariance matrix. By noise-normalizing the spectra prior to the application of the PCA, the ability to fit the data is enhanced by avoiding giving too much weight to variance caused by noise. Giving m\* the number of most significant eigenvectors of $S_\epsilon$, one can represent the spectrum in the eigenspace by a truncated vector of principal component scores, p\* of rank m\* (m\* < m). p\* is thus a compressed representation of y. The reconstructed spectrum, $\widetilde{y}$ (dimension m) is given by:*

$$\widetilde{y} = \overline{y} + NE^*p^* \qquad (3)$$

*where $E^*$ is the matrix of the m\* first eigenvectors or principal components. We define the noise normalized residual vector r (dimension m) of the reconstruction by:*

$$r = N^{-1}(y - \widetilde{y}) \qquad (4)$$

*By definition, if m\* is taken equal to m, $\widetilde{y} = y$ and the residual is the null vector. In nominal cases if the truncation rank is carefully chosen, r essentially contains noise. Several techniques exist to estimate m\* in order to keep the essential part of the atmospheric signal and to remove the eigenvectors containing mainly the measurement noise (e.g., Antonelli et al. (2004), Atkinson et al. (2010)).*
*In the following the noise normalized residual, which is calculated for each IASI IFOV, is called IFOV-residual.*

The small size (120000) of the training database is problematic because the computation of the 8461\*8461 covariance matrix will be affected by instrument noise as well as unusual spectra (which can be hard to avoid).

We added in Section 3.2 some explanations on the size of the training database, as well as some additional references:

*The training set includes spectra observed over different types of atmospheric/surface conditions at different scan angles and for different pixel numbers to ensure that a truncated set of eigenvectors can be adequately used to represent any observed spectrum. Additionally, if the training set is too small, the specific outcome of the random noise will not be sufficiently uncorrelated and uniform, and will therefore have an influence on the computed eigenvectors and eigenvalues. Extensive experience on IASI spectra from EUMETSAT (Hultberg, 2009, https://www.eumetsat.int/media/8306) and additional experiments with different dataset sizes show that a number of about 70000 spectra is a reasonable lower limit.*

Hultberg, T.: IASI Principal Component Compression (IASI PCC) FAQ, March 2009, EUMETSAT technical note, available at https://www.eumetsat.int/media/8306, 2009.

There is no evidence presented for the usefulness of the thinning towards the poles "in order to not over-represent high latitudes".

For a sake of clarity, the sentence has been changed to:
*To avoid over-representing high latitudes, because of the large swath of IASI (~2200 km) and frequent overpasses over this area with the polar orbiting satellites, the following method was applied:*

Line 131: then? And how can a random selection help to "represent all the conditions" – should be better to keep all.

This comment was taken into account and the following paragraphs were added in Section 3.2 of the revised manuscript:

*To avoid over-representing high latitudes, because of the large swath of IASI (~2200 km) and frequent overpasses over this area with the polar orbiting satellites, the following method was applied:*

*To reach a sufficient but reasonable number of IASI spectra/IFOVs (1.3 $10^6$ spectra per day, 4.7 $10^8$ per year), 120000 IFOVs for year 2013 were randomly chosen to represent all atmospheric/surface situations (air masses, land/sea, day/night, clear/cloudy) and acquisition conditions (IASI scan mirror position and pixel number).*

The reference to EUMETSAT extensive work with IASI to build the training database based on random selection has also been added to this Section 3.2 (Hultberg, 2009).

Hultberg, T.: IASI Principal Component Compression (IASI PCC) FAQ, March 2009, EUMETSAT technical note, available at https://www.eumetsat.int/media/8306, 2009.

Section 3.3: 20? Actually 1 PC is enough to "depict (sic!) most of the atmospheric variability". While 150 PC is a good choice this is not related to any percentage of the total variance but rather the signal to noise ratio of the remaining PCs.

Section 3.3. has been improved and additional information has been provided to explain and justify the choice for the truncation as follows:

*Several techniques exist to estimate m\* in order to keep the essential part of the atmospheric signal and to remove the eigenvectors containing mainly the measurement noise. Antonelli et al (2004) define a criterium based on the spectral RMS reconstruction residuals, finding the optimal truncation rank when this value approach the spectral RMS of the instrument noise. Other methods test directly the behavior of the reconstruction score* $\sqrt{\frac{1}{m}\sum_{i=1}^{m} r_i^2}$ *as a function of the truncation rank, by looking at the second derivative of the reconstruction score as a function of the truncation rank (e.g., Hultberg, 2009) or by plotting the principal reconstruction score spatial correlation as a function of eigenvector rank (Atkinson et al., 2009). In this study, the choice of is based on the analysis of the eigenvalues. They quantify the variability explained by the corresponding eigenvectors, and the optimal number of eigenvectors needed to reproduce the signal in the raw radiances can be determined by analyzing the magnitude and behavior of the eigenvalues (sorted in descending order). In the present implementation of the PCA method we process the full IASI spectrum and use a simple method for selecting the truncation rank. The plot of the eigenvalues was examined and PCs were selected up to the point where the slope of the curve stabilized. This leads to choose the first 150 eigenvectors as done in Atkinson et al., 2010. Sensitivity tests has been performed to test the impact of using different values (from 120 to 250) on the reconstructed scores obtained on several atmospheric events (fires and volcanoes cases discussed in the next sections) and confirm this value.*

Section 4.2 has been extensively modified and rewritten to improve the detection threshold definition as follows:

**4.2 Detection thresholds**

[revised manuscript text omitted]

Why not apply thresholds for each individual spectrum, instead of the granule min and max? Faster (line 152)? No, I don't believe so.

Section 4.2 was rewritten and only two thresholds are applied: a first threshold based on the granule in order to select the granules associated with outliers only (which allows to gain computation time). A 2$^{nd}$ threshold is applied on each individual residual in order to select the residual associated with reconstruction errors. See above for the revised Section 4.2.

The sentence line 152 was removed.

The use of 3 different thresholds seems unnecessarily complicated. And the second threshold might be counterproductive in case most of the granule is affected by a similar anomaly.

It is true the second threshold might be counterproductive in case most of the granule is affected by a similar anomaly. However, this case is very rare.

The $F_2$ threshold was defined to detect outliers that we were not able to associate with species for research aspects. This part of this work is not described in the manuscript and will be the subject of future work. Since the $F_2$ threshold is not used in this manuscript, it was removed. See above for the revised Section 4.2.

The term "signal intensity" is used without being introduced. It is simply the (absolute value of) the (noise normalised) residual and the new term does only confuse.

We changed "signal intensity" by "noise normalized residual" everywhere.

It is not clear why a second training set was used for the computation of thresholds. Why different threshold for day and night?

The training data defined for the calculation of the principal components was generated from spectra chosen randomly and associated with no extreme events. Contrarily, the dataset defined for the calculation of the thresholds was generated from granules (because the IASI-PC-GE method is based on granules) chosen randomly and associated with all atmospheric situations, including extreme event conditions. Because of this, a second training set had to be generated for the definition of thresholds. The following sentence was added in the revised manuscript:

*Note that this dataset differs from that generated for the principal component calculation as the detection method is applied on a granule basis.*

The IASI signal significantly differs for day- and night-time conditions, which is mainly due to thermal contrast. This directly affects the reconstruction residual amplitude, and thus IASI-PCA sensitivity is generally lower during night-time than during day-time. Using the same threshold for day and night-time would lead to fewer detection during night-time. This is the main reason for using different thresholds for day and night conditions.

We added the following sentence in the revised manuscript:

*Since IASI-PCA sensitivity is generally lower during night-time than during day-time, which is mainly due to thermal contrast, different thresholds for day and night conditions were defined.*

Line 442-443 and 506-507: This is what was already discussed on page 10. I feel it is wrong to talk about "artefact" and "reconstruction error" in this context. The detection method of the paper is nothing but the identification of reconstruction error. That an unusual perturbation in a limited number of channels can affect the reconstruction residual in other channels is natural and unavoidable (as you can convince yourself by looking at a two-dimensional space with 1 retained PC). Actually maybe the

biggest contribution of the paper is that is shows that this kind of "cross talk" seems to be relatively rare in practice.

We agree and would like to thank the referee for his comment. The text has been modified accordingly in Section 5.3.2 and Section 6 as follows:

*The detection at ~1326 cm$^{-1}$ is not associated to HNO$_3$ and is due to the contribution of SO$_2$ and aerosols, as already discussed in the case of Ubinas eruption (see section 5.1.1).*

*Finally, as explained above concerning SO$_2$ and HNO$_3$, the spectral coincidence of some of the intense spectral features of these two species can affect the reconstruction of one when the other one is highly present. In the frame of this study, this is the only identified example of confounding situations (i.e., unusual perturbation in a limited number of channels impacts the reconstruction residual in other channels) leading to false detection. Considering the high numbers and diversity of detections and extreme situations analyzed in this work, such confounding situations are rare and PCA-based detection of atmospheric events can be effectively and efficiently exploited.*

Technical corrections

Line 19: horizontal => spatial

Done

Line 120: insert "as"

Done

Line 145: "maximum of information"?

This sentence was removed.

Line 154: noise normalised "residuals"

This sentence was removed.

Line 182: I don't understand this sentence

The sentence was removed.

---

## Author Comment (AC2)

**Response to Referee #1**

**The authors would like to thank the referee for her/his review. Below are our responses to the comments brought up by the referee. Referee's comments and our replies are marked in blue and in black, respectively. In italic are the changes made in the manuscript.**

The manuscript describes a PCA based method on real time detection and characterization of atmospheric events. For this, they are applying measured data from three IASI satellites. The manuscript is nicely written and offers interesting application of the PCA on detection of extreme events. However, some clarifications are needed, as described below.

Major comment
The methodology description needs to be improved. Even though majority of the methodology is described in previous study, it would be important to provide here necessary details on the method for replicating the analyses with similar data. e.g. PCA could be described more clearly and it is not clear how to you get GMI and GMA from the PC's. This makes it more difficult to follow the results from the case studies.

Section 3 has been extensively modified and rewritten to improve the PCA methodology description, See the revised Section 3 at the end of this document in Appendix A.

The explanation to obtain the GMI/GMA was clarified in Section 4.1 as follows:

*Each granule contains ~2700 radiance spectra, from which the corresponding IFOV-residuals are computed based on the IASI-PCA method. For each granule, the largest positive and negative residual value for each spectral channel is recorded in two arrays, called hereafter "Granule Maxima" (GMA) and "Granule Minima" (GMI). GMI and GMA are defined as pseudo-residuals of dimension 8461 (the number of radiance channels) and represent the spectral envelope of the statistics of residuals over the granule. Physically, the GMI (GMA) pseudo-residual is associated with reconstruction errors of spectral absorption (emission) lines. Since the method is based on the granule extrema (GMI and GMA), the method is therefore called: IASI-PCA-GE, with GE standing for Granule-Extrema. It is important to note that these pseudo-residuals associated with a granule are different from the individual IFOV-residual associated with each IFOV.*

Specific comments
Abstract: Point out the focus and true novelty of this manuscript in the abstract. Now it sounds more like the introduction

As suggested by the Referee, we added a paragraph pointing out the focus and true novelty of the manuscript in the abstract:

*The method is running continuously, delivering email alerts on a routine basis using the near real time IASI L1C radiance data. It is planned to be used as an online tool for the early and automatic detection of extreme events, which was not done before.*

line 117, Antonelli 2004 is not in the list of references. In addition, with Atkinson 2008 and 2010 they are not the original or the best references of methods for defining the optimal number of components

We thank the referee. Antonelli (2004) was added to the list of references. In addition, this sentence and corresponding references have been consolidated and moved to Section 3.3, see our answer to the corresponding comment below.

Line 173: Define IASI-PCA-GE more clearly
We added the following sentence in Section 4.1:

*Since the method is based on the granule extrema (GMI and GMA), in the following the method is called: IASI-PCA-GE, with GE standing for Granule-Extrema.*

Section 3.3.: As the explained variance is not really increasing after ~25 components, using 150 PC sounds a bit of overfitting. How did you define the number? How many PC would e.g. Scree test or Kaiser criterion suggest?

Section 3.3 has been consolidated to better explain and justify the choice of 150 PC. Additional references have been also added (Hultberg, 2009, Atkinson, 2009). Note however that :

- We chose to provide only references related to the use of PCA on IASI, as there is already a large experience with the PCA on these specific instruments, thanks to the work performed at EUMETSAT for defining and operationnaly implementing the Principal Component Compression for the IASI L1D products, and to the scientific work performed for testing and analysing this processing and the outliers. As already explained in these references, different approaches can be used for the choice of the truncation. One important aspect is that the performance of the reconstruction depends (in a complex and correlated manner) on several choices : the training dataset, the normalisation matrix, the truncation threshold. For these choices, the specific experience gained on IASI is critical.
- At the end the "optimal" choice remains empirical and statistically-based. Sensitivity tests on the different parameters (including the choice of the truncation) is thus a key point. This is now mentioned in the revised manuscript. It is of particular importance in our work, as the objective here is not to perform the best reconstruction of all the measurements, but to detect outliers.

Atkinson, N. C., Ponsard, C., and Hultberg, T.: AAPP enhancements for the EARS-IASI service, Proc. EUMETSAT Meteorological Satellite Conf., Bath, UK, 21–25 September 2009, available at https://www-cdn-int.eumetsat.int/files/2020-04/pdf_conf_p55_s8_39_atkinson_p.pdf, 2009.

Hultberg, T.: IASI Principal Component Compression (IASI PCC) FAQ, March 2009, EUMETSAT technical note, available at https://www.eumetsat.int/media/8306, 2009.

See the revised Section 3.3 at the end of this document in Appendix A.

Lines 501-506: With this high number of observations in the training set, it is not probable that few outliers would affect drastically to the sensitivity of the method. As already the Atkinson papers pointed out, there has been suspicions that PCA might not be the best method for this type of analysis. Have you considered other possible factorization methods like EFA, NMF or PMF discussed e.g. in Isokäänta et al. 2020 (https://doi.org/10.5194/amt-13-2995-2020)?
In addition, have you considered accounting for the geophysical parameter possibly acting as confounding factors in your analysis?

We agree that it was not correct to write that few outliers would affect drastically to the sensitivity of the method. To improve and clarify the discussion, the corresponding sentences have been modified in the revised manuscript. In particular, the main argument explaining the unconclusive results on CO has been added, in agreement with your comment:

*Also, unconclusive results were obtained for CO because its variability is already well captured by a truncated reconstruction due to the high variability of this species, from background conditions (50 ppb) to highly polluted areas (4000 ppb).*

However, other possible methods have not been tested in this work, as the choice of testing PCA analysis of IASI measurements for extreme event detection is at the origin of the presented work. Finally, we considered accounting for the geophysical parameters acting as confounding factors, as it is illustrated for the $HNO_3$ species in the discussion of the The Ubinas case in Section 5.1.1. a sentence was also added in the revised manuscript following your remark :

*Finally, as explained above concerning SO₂ and HNO₃, the spectral coincidence of some of the intense spectral features of these two species can affect the reconstruction of one when the other one is highly present. In the frame of this study, this is the only identified example of confounding situations (i.e., unusual perturbation in a limited number of channels impacts the reconstruction residual in other channels) leading to false detection.*

Conclusions: point out that the method can be used as online tool for detecting extreme events, as mentioned in the text earlier.

We thank the Referee for his suggestion. We added the following paragraph in Section 6:

[revised manuscript text omitted]